# Comparative Metabolomic Profiling of L-Histidine and NEFA Treatments in Bovine Mammary Epithelial Cells

**DOI:** 10.3390/ani14071045

**Published:** 2024-03-29

**Authors:** Wenqiang Sun, Mengze Li, Hanjun Ren, Yang Chen, Wei Zeng, Xiong Tan, Xianbo Jia, Shiyi Chen, Jie Wang, Songjia Lai

**Affiliations:** 1State Key Laboratory of Swine and Poultry Breeding Industry, College of Animal Science and Technology, Sichuan Agricultural University, Chengdu 611130, China; wqsun2021@163.com (W.S.); 15706065790@163.com (M.L.); 17392547544@163.com (H.R.); chenyi154121@163.com (Y.C.); jaxb369@sicau.edu.cn (X.J.); chensysau@163.com (S.C.); wjie68@163.com (J.W.); 2Key Laboratory of Livestock and Poultry Multi-omics, Ministry of Agriculture and Rural Affairs, College of Animal Science and Technology, Sichuan Agricultural University, Chengdu 611130, China; 3Farm Animal Genetic Resources Exploration and Innovation Key Laboratory of Sichuan Province, Sichuan Agricultural University, Chengdu 611130, China; 4Sichuan Province Animal Husbandry Science Research Institute (Yangping Breeding Bull Farm), Meishan 620360, China; hy7408363@163.com (W.Z.); 15984302711@163.com (X.T.)

**Keywords:** NEFA, L-histidine, untargeted metabolomic analysis, cattle, BMECs

## Abstract

**Simple Summary:**

Ketosis in cows occurs when excessive ketone bodies are produced by the liver, typically due to energy deficits from reduced feed intake or specific dietary conditions. Scientists have noticed that cows with ketosis have lower amounts of a substance named L-histidine. This sparked curiosity about how L-histidine might help cows with this problem. To understand this better, researchers looked at changes in cow mammary cells caused by the fat and whether L-histidine could help. They found many changes in the cell’s substances, some increasing and some decreasing, showing that the cells were trying to respond to the fat. Interestingly, when L-histidine was added, it seemed to help by changing some of these substances, suggesting it might protect the cells. One particular substance seemed to play a big role and was affected by both the fat and L-histidine in different ways. This research helps us understand the complicated ways cow cells react to excess fat and suggests that L-histidine could be a helpful tool in keeping cows healthy, which is valuable for farmers and the dairy industry.

**Abstract:**

Non-esterified fatty acids (NEFAs) are pivotal in energy metabolism, yet high concentrations can lead to ketosis, a common metabolic disorder in cattle. Our laboratory observed lower levels of L-histidine in cattle suffering from ketosis, indicating a potential interaction between L-histidine and NEFA metabolism. This relationship prompted us to investigate the metabolomic alterations in bovine mammary epithelial cells (BMECs) induced by elevated NEFA levels and to explore L-histidine’s potential mitigating effects. Our untargeted metabolomic analysis revealed 893 and 160 metabolite changes in positive and negative models, respectively, with VIP scores greater than 1 and *p*-values below 0.05. Notable metabolites like 9,10-epoxy-12-octadecenoic acid were upregulated, while 9-Ethylguanine was downregulated. A pathway analysis suggested disruptions in fatty acid and steroid biosynthesis pathways. Furthermore, L-histidine treatment altered 61 metabolites in the positive model and 34 in the negative model, with implications for similar pathways affected by NEFA. Overlaying differential metabolites from both conditions uncovered a potential key mediator, 1-Linoleoylglycerophosphocholine, which was regulated in opposite directions by NEFA and L-histidine. Our study uncovered that both NEFA L- and histidine metabolomics analyses pinpoint similar lipid biosynthesis pathways, with 1-Linoleoylglycerophosphocholine emerging as a potential key metabolite mediating their interaction, a discovery that may offer insights for therapeutic strategies in metabolic diseases.

## 1. Introduction

Non-esterified fatty acids (NEFAs) play a crucial role in energy metabolism. Elevated levels of NEFAs can lead to the production of ketones, a process associated with the onset of ketosis [1]. However, excessive NEFA concentrations can have adverse effects on health. High NEFA concentrations upregulate acyl-CoA synthetase long-chain members (ACSL) and acyl-CoA dehydrogenase long-chain members (ACADL), enhance carnitine palmitoyltransferase 1A (CPT IA) expression up to a certain point, and decrease acetyl-CoA carboxylase (ACC) expression, suggesting a facilitation of fatty acid activation and β-oxidation but potentially hindering mitochondrial translocation at excessive levels [2]. Such increased NEFA levels are also significantly linked to a greater risk of cardiovascular disease in nondiabetic individuals with a type 2 diabetic family history, highlighting NEFA’s role in cardiovascular pathogenesis [3]. In dairy cows, elevated NEFAs induce oxidative stress, leading to an imbalance between oxidants and antioxidants, the activation of apoptotic pathways, and hepatocyte apoptosis [4]. Additionally, high NEFA levels can initiate proinflammatory signaling through interactions with TLRs, causing the release of cytokines like IL-6 [5], and induce ER stress that activates UPR and inflammatory pathways [6]. Furthermore, NEFAs have been implicated in affecting cell growth by inducing apoptosis via the PI3K/AKT/FoxO1 pathway [7]. These insights underscore the complex and potentially harmful effects of high NEFA levels on health, emphasizing the need for careful management of NEFA concentrations in metabolic disorders.

In ketotic cattle, L-histidine levels are lower compared to healthy counterparts, highlighting its importance; L-histidine is crucial for various proteins, and its deficiency may result in reduced body weight [8]. Supplementation with L-histidine could potentially improve insulin sensitivity and reduce obesity-related parameters such as BMI, fat mass, and NEFA levels while also mitigating inflammation and oxidative stress in individuals with metabolic syndrome, possibly by inhibiting the NF-κB signaling pathway in adipocytes [9]. In the context of dairy farming, L-histidine, along with sodium acetate, can boost β-casein production in nutrient-restricted bovine mammary cells, indicating that L-histidine alone can be beneficial even without additional energy sources [10]. This amino acid also appears to enhance growth performance in fish [11] and can suppress inflammatory cytokines like IL-6 and TNF-a in LPS-induced macrophages [12], as well as increase antioxidant enzyme activities such as GPX and SOD [13], suggesting a broader role for L-histidine in counteracting the adverse effects of high NEFA levels.

Therefore, the intricate interplay between NEFA and L-histidine in bovine health is of significant interest, particularly due to the role of NEFA in energy metabolism and its association with ketosis when at high concentrations. Our preliminary findings have identified a decrease in L-histidine levels in cattle with ketosis [14], suggesting a potential regulatory relationship between L-histidine and NEFA metabolism. This study aims to delve deeper into the metabolomic responses of BMECs to high NEFA levels and to assess the potential of L-histidine in counteracting these effects. Through comprehensive metabolomic profiling, we have pinpointed key metabolic alterations and pathways influenced by NEFA and L-histidine, proposing 1-Linoleoylglycerophosphocholine as a potential mediator. The insights gained from this research could lead to novel strategies for managing ketosis, improving cattle health and productivity through targeted metabolic interventions.

## 2. Materials and Methods

Cell culture and Treatments

The bovine mammary epithelial cell line (BMECs, MAC-T) was propagated in DMEM/F12 medium enriched with 10% fetal bovine serum (FBS, Gibco, Carlsbad, CA, USA). Cultivation was conducted in 25 cm^2^ Corning culture flasks (Corning, NY, USA). Upon achieving 85–90% confluence, cells were seeded into 96-well or 6-well plates (Corning, NY, USA) for subsequent treatments. L-histidine exposures were carried out using concentrations of 0, 0.2, 0.4, and 0.8 mM L-histidine (Sigma-Aldrich, St. Louis, MO, USA) over a period of 24 h. NEFA treatments were similarly conducted with concentrations of 0, 0.2, 0.4, and 0.6 mM NEFA for 24 h. Following treatment, cells were either subjected to the CCK-8 assay for viability analysis or harvested for metabolomic assessments. The 10 mmol/L stock solution of NEFA contained 0.53 mmol/L of palmitoleic acid, 0.49 mmol/L of linoleic acid, 4.35 mmol/L of oleic acid, 1.44 mmol/L of stearic acid, and 3.19 mmol/L of palmitic acid [15].

CCK-8 assay

BMECs were seeded into 96-well plates and grown until they reached 70% confluence, at which point they were treated with varying concentrations of NEFA and L-histidine. For the detection, 10 µL of CCK-8 reagent (Beyotime, Shanghai, China) was added to each well, followed by a 2 h incubation at 37 °C in a CO_2_ incubator. The absorbance of the samples at 450 nm was then measured using a Varioskan LUX multimode microplate reader (Thermo Scientific, Waltham, MA, USA).

Metabolite Extraction

The metabolomics analysis utilizes an integrated LC/MS system comprising Waters Acquity I-Class PLUS ultra-high performance liquid chromatography (UHPLC) coupled with a Waters Xevo G2-XS QTOF high-resolution mass spectrometer. Analytical separation is achieved using a Waters Acquity UPLC HSS T3 column (1.8 µm, 2.1 × 100 mm). For the positive ionization mode, the mobile phase A consists of a 0.1% formic acid solution in water, and the mobile phase B is 0.1% formic acid in acetonitrile. Similarly, for the negative ionization mode, mobile phase A is a 0.1% formic acid solution in water, with mobile phase B being 0.1% formic acid in acetonitrile.

LC-MS/MS Analysis

The Waters Xevo G2-XS QTOF high-resolution mass spectrometer is capable of the simultaneous acquisition of both primary and secondary mass spectrometric data in MSe mode, managed by the MassLynx V4.2 software from Waters. It conducts dual-channel data collection within each cycle, capturing spectra at both low collision energy (set at 2 V) and a variable high collision energy range (spanning from 10 V to 40 V), with a spectral acquisition rate of 0.2 s per scan. For the chromatographic separation, the mobile phase in positive ion mode consists of 0.1% formic acid aqueous solution (mobile phase A) and 0.1% formic acid in acetonitrile (mobile phase B); similarly, in the negative ion mode, it uses the same mobile phase A and B composition. The parameters for the ESI (electrospray ionization) source are set as follows: the capillary voltage is configured at 2000 V for the positive ion mode and −1500 V for the negative ion mode; the cone voltage is maintained at 30 V. The temperature of the ion source is controlled at 150 °C, while the desolvation gas is heated to 500 °C. The backflush gas flow is regulated at 50 L/h and the desolvation gas flow at 800 L/h.

Data preprocessing and annotation

The raw spectral data, acquired via MassLynx V4.2, undergoes preprocessing with Progenesis QI software V3.0, which facilitates peak detection, alignment, and additional data processing tasks. Compound identification is performed by referencing the online METLIN database within the Progenesis QI software, as well as utilizing Biomark’s proprietary library. Concurrently, theoretical fragmentation patterns and mass accuracy are meticulously validated, ensuring all mass deviations remain within a stringent threshold of 100 ppm.

Data analysis

The preprocessing step involves normalizing the raw peak area data against the total peak area to ensure consistency before proceeding with further analysis. To assess the reproducibility of sample data within groups, including quality control samples, we employ principal component analysis (PCA) and Spearman’s correlation analysis. For compound identification, we query classification and pathway information against databases such as KEGG [16] and HMDB [17]. We then calculate the fold changes in identified compounds based on group classifications and employ a *t*-test to determine the statistical significance (*p*-value) of variations observed in each compound. The R package ‘ropls’ is utilized for orthogonal projections to latent structures–discriminant analysis (OPLS-DA) modeling. To ensure the model’s robustness, we conduct 200 permutation tests. The model’s variable importance in projection (VIP) scores are computed through multiple cross-validations. The screening criteria applied are a *p*-value < 0.05 and a VIP score > 1. Finally, we perform a hypergeometric distribution test to determine the significance of enriched KEGG pathways among the differentiated metabolites.

## 3. Results

Effect of High NEFA Levels on the Metabolomics of BMECs in a Positive Model

To investigate the impact of high levels of NEFA on the metabolomic profile of BMECs, we conducted an untargeted metabolomic analysis. We focused on endogenous substances with a variable importance in projection (VIP) score greater than 1 and a *p*-value less than 0.05 to assess the metabolic alterations induced by high NEFA levels in BMECs. According to Appendix A, 893 metabolites met these criteria in the positive model BMECs, with 802 metabolites showing increased levels and 91 showing decreased levels compared to the control BMECs. These metabolites were depicted in a volcano plot (Figure 1A) and a heatmap (Figure 1B) for visual comparison. The three most upregulated metabolites were 9,10-epoxy-12-octadecenoic acid, arbaprostil, and (4Z,7Z,11Z,13Z,16Z,19Z)-10-Hydroxydocosa-4,7,11,13,16,19-hexaenoylcarnitine, whereas the top three downregulated metabolites included 9-Ethylguanine, N2-(1-Carboxyethyl)-2′-deoxyguanosine, and 3-ketosphingosine (Figure 1C). These differential metabolites were annotated using the KEGG database, revealing enrichment in pathways such as fatty acid biosynthesis, steroid biosynthesis, and the biosynthesis of unsaturated fatty acids, as illustrated in Figure 1D. Additionally, we performed significance testing for metabolite correlation analysis (Figure 1E).

Effect of High NEFA Levels on the Metabolomics of BMECs in a Negative Model

Beyond the positive model, Appendix A indicates that in BMECs exposed to high NEFA levels, 160 endogenous metabolites were identified with a VIP score greater than 1 and a *p*-value below 0.05. Among these, 70 metabolites were found in higher concentrations, while 90 were in lower concentrations compared to the control BMECs. The distribution of these 160 metabolites is depicted in a volcano plot (Figure 2A) and a heatmap (Figure 2B). The three most significantly upregulated metabolites include 3–Pentadecenal, avocadene, and (–)–alpha–Bisabolol, whereas the top three downregulated metabolites include 4–Methylthiobutyl–desulfoglucosinolate, abscisic aldehyde, and methyl beta–D–glucopyranoside (Figure 2C). These differential metabolites have been cataloged in the KEGG metabolite database, which highlighted an enrichment in cholesterol metabolism and steroid hormone biosynthesis pathways, as shown in Figure 2D. Furthermore, a significance test was conducted for metabolite correlation analysis (Figure 2E).

Effect of L-histidine on the Metabolomics of BMECs in a Positive Model

Based on previous studies, cows with ketosis, who typically exhibit high levels of NEFA, have lower levels of L-histidine compared to healthy cows [14]. To investigate the effect of L-histidine on the metabolomic profile of BMECs, we performed an untargeted metabolomic analysis. We focused on endogenous substances with a VIP score greater than 1 and a *p*-value less than 0.05 to assess the metabolic alterations induced by L-histidine in BMECs. According to Appendix A, 61 metabolites met these criteria in the positive model BMECs, with 29 metabolites showing increased levels and 32 showing decreased levels compared to the control BMECs. These metabolites were depicted in a volcano plot (Figure 3A) and a heatmap (Figure 3B) for visual comparison. The three most upregulated metabolites were sphingosine, DG(20:0/20:5(5Z,8Z,11Z,14Z,16E)–OH(18R)/0:0), and L–histidine, whereas the top three downregulated metabolites included Yuzu lactone, 2,4–Dimethyl–2E,4E–hexadien–1–ol, and 24–Acetyl–25–cinnamoylvulgaroside (Figure 3C). These differential metabolites were annotated using the KEGG database, revealing enrichment in pathways such as fatty acid biosynthesis, steroid biosynthesis, and the biosynthesis of unsaturated fatty acids, as illustrated in Figure 3D. Additionally, we performed significance testing for metabolite correlation analysis (Figure 3E).

Effect of L-histidine on the Metabolomics of BMECs in a Negative Model

Beyond the positive model, Appendix A indicates that in the BMECs exposed to L-histidine, 34 endogenous metabolites were identified with a VIP score greater than 1 and a *p*-value below 0.05. Among these, 16 metabolites were found in higher concentrations, while 18 were in lower concentrations compared to the control BMECs. The distribution of these 34 metabolites is depicted in a volcano plot (Figure 4A) and a heatmap (Figure 4B). The three most significantly upregulated metabolites include 3–{[(2E)–4–Amino–4–oxobut–2–enoyl]amino–L–alanine, PE(18:1(11Z)/20:4(5Z,8Z,11Z,14Z)) and L–Aspartate, whereas the top three downregulated metabolites include momorcharaside B, Isoleucyl–prolyl–proline and Mebeverine alcohol (Figure 4C). These differential metabolites have been catalogued in the KEGG metabolite database, which highlighted an enrichment in cholesterol metabolism and steroid hormone biosynthesis pathways, as shown in Figure 4D. Furthermore, a significance test was conducted for metabolite correlation analysis (Figure 4E).

1-Linoleoylglycerophosphocholine Could be the Pivotal Mediator Linking NEFA and Histidine

To investigate the potential mediator that connects NEFA and histidine, we overlaid the differential metabolites from both NEFA and histidine across positive and negative models. Figure 5A,B illustrate that five differential metabolites were identified in the positive model, while only one was found in the negative model upon overlapping. Intriguingly, 1–Linoleoylglycerophosphocholine levels increased following NEFA treatment but decreased after L-histidine treatment (Figure 5C). This finding suggests that 1–Linoleoylglycerophosphocholine may serve as a crucial intermediary between NEFA and histidine.

## 4. Discussion

Elevated NEFA levels, key serum metabolites, are implicated in various detrimental health outcomes. Studies have linked high NEFA concentrations to an augmented familial risk of cardiovascular disease [3] and have demonstrated their role in disrupting nitric oxide-independent vasodilation [18], as well as impairing oocyte maturation and early embryo development, which are critical for fertility [19]. Notably, type 2 diabetes has been correlated with increased NEFA levels, often reaching 0.6–2.4 mM, which is attributed to defective NEFA clearance in adipose tissue rather than enhanced lipolysis, with glycemic control and visceral fat being significant contributors [20]. NEFAs are also being investigated as potential biomarkers for the diagnosis of depression in adolescents [21]. On a molecular level, high NEFA concentrations counteract the regulatory effects of hyperglycemia on endogenous glucose production by influencing hepatic glycogenolysis under steady hormonal conditions [22] and induce lipotoxicity in hepatocytes, as evidenced by reduced cell viability, altered lipid gene expression, and disrupted metabolic pathways crucial for lipid management [23]. Moreover, our study has found that BMECs subjected to elevated NEFAs exhibit a considerable alteration in 893 metabolites, predominantly those associated with fatty acid and steroid biosynthesis pathways. In contrast, BMECs under high NEFA stress show significant shifts in 160 metabolites, particularly within cholesterol and steroid hormone metabolism.

Transitioning into lactation, dairy cows often experience a negative energy balance, leading to increased NEFAs [24]. Elevated levels of NEFA can lead to the production of ketones, a process associated with the onset of ketosis [1]. Previously, our lab found that lower L-histidine levels in ketotic cattle compared to healthy ones underscore its significance [8]. L-histidine is a vital amino acid involved in proton buffering, metal ion chelation, antioxidant activity, erythropoiesis, and histamine-related processes [25]. Histidine supplementation in rats modifies protein and amino acid metabolism, increasing body weight, liver, and kidney protein levels, plasma amino acids and ammonia, muscle alanine and glutamine, and proteolysis, while influencing the balance of histidine-containing peptides and the requirements for methionine and glycine [26]. Increased plasma histidine activates hepatic STAT3, suppressing hepatic glucose production (HGP) in type 2 diabetes by inhibiting gluconeogenic enzymes and enhancing insulin’s effect, an action independent of central insulin signaling but reliant on central histamine pathways, revealing a novel target for diabetes treatment [27]. However, high-dose histidine administration during cardiac surgery for myocardial protection can cause elevated ammonia levels, altered amino acid profiles, and disrupted energy metabolism across tissues, indicating a broad impact on biochemical equilibrium [28]. On a molecular level, histidine supplementation restructured liver metabolism, enhancing antioxidant defenses and nitric oxide synthesis while elevating key amino acids and glutathione levels [29]. In our study, the positive model BMECs analysis revealed 61 metabolites with a split of 29 increased and 32 decreased compared to the controls, affecting crucial pathways such as fatty acid and steroid biosynthesis according to the KEGG database, as illustrated by visual aids and supported by correlation analysis. In the negative model with L-histidine exposure, 34 metabolites were significantly altered—16 higher and 18 lower—with significant shifts in amino acids like L-Aspartate and certain peptides, and this was linked to biosynthesis pathways and confirmed by correlation analysis.

To investigate the intermediary that connects NEFA and histidine, we analyzed the differential metabolites associated with both substances using positive and negative models, identifying five in the former and one in the latter. Remarkably, levels of 1–Linoleoylglycerophosphocholine increased post–NEFA treatment but decreased after histidine was administered. Previous metabolomic screenings of fasting plasma from non–diabetic individuals pinpointed α–hydroxybutyrate (α–HB) and linoleoylglycerophosphocholine (LGPC) as combined indicators of insulin resistance (IR) and glucose intolerance [30]. LGPC showed no link to obesity or lipid markers, yet it was significantly and inversely associated with glucose clearance, with elevated levels resulting in heightened post-challenge glucose in non-diabetics, highlighting its potential role as a marker for IR and impaired glucose metabolism [31]. AHB, LGPC, oleic acid, and insulin have been utilized as the foundation for an insulin resistance assay since all four are recognized as biomarkers of insulin sensitivity [32]. Moreover, malnutrition-induced insulin resistance in ruminants is marked by lowered glucose levels and impaired insulin and glucose dynamics, along with an increase in lipid mobilization and ketogenesis—this is particularly noticeable in ketotic cows, which exhibit diminished insulin response and heightened tissue insulin resistance [33], lending further credence to the significance of LGPC. Consequently, these insights suggest that 1-Linoleoylglycerophosphocholine may serve as a pivotal intermediary between NEFA and histidine.

The clinical implications of our research are profound for the dairy industry as the metabolic interplay between NEFA and L-histidine in bovine mammary epithelial cells, particularly the role of 1-Linoleoylglycerophosphocholine, suggests a promising avenue for proactive herd health management. By incorporating this metabolic testing into routine herd management, cattle farmers can proactively address energy imbalances, potentially enhancing milk production and overall herd health. While our primary goal is to advance scientific knowledge in this area, we recognize the potential translational benefits of our research. Accordingly, we have begun a thorough investigation into the commercial viability of these findings. This includes considering intellectual property rights and patent filing as necessary steps toward commercial development. Our prudent approach ensures that we maintain scientific integrity while also considering the practical applications that could significantly benefit dairy industry practices.

## 5. Conclusions

In summary, our study discovered that high NEFA levels induce significant disruptions in the metabolic processes of BMECs, notably affecting fatty acid and steroid biosynthesis pathways. L-histidine treatment shows a promising modulatory effect, altering metabolites and suggesting a partial rebalancing of disrupted pathways. The differential regulation of the mediator 1-Linoleoylglycerophosphocholine by NEFA and L-histidine underscores its potential as a therapeutic target. This study suggests that L-histidine could be a viable intervention to mitigate NEFA-induced metabolic challenges in cattle, with implications for improving bovine health.

## Figures and Tables

**Figure 1 animals-14-01045-f001:**
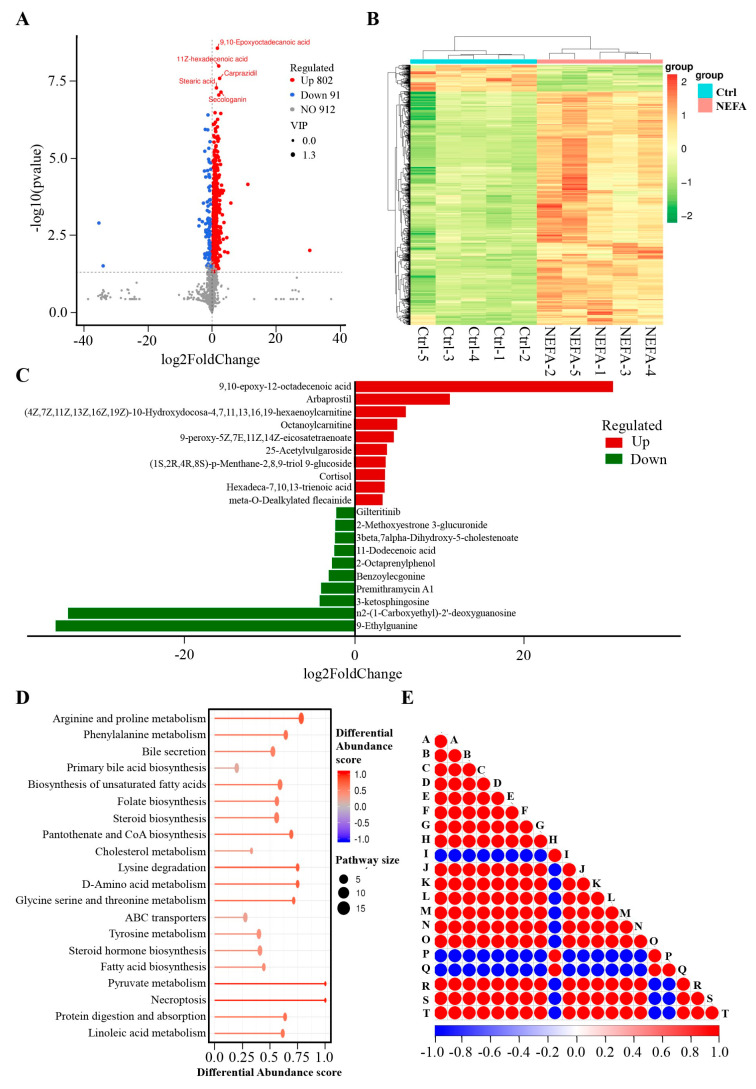
Effect of high NEFA levels on the metabolomics of BEMCs in the positive model. (**A**) Volcano of metabolites with significant differences between control and NEFA treatment groups. The criteria for upregulated differential metabolites should be a VIP > 1, *p* < 0.05, log2(Foldchange) > 0. Conversely, the criteria for downregulated differential metabolites should be a VIP > 1, *p* < 0.05, and log2(Foldchange) < 0. (**B**) The clustering heat map of differential metabolites between control and NEFA treatment groups. (**C**) Top 10 elevated and reduced differential metabolites identified between the control and NEFA groups in the positive model (**D**) KEGG pathways enriched by differential metabolites between control and NEFA treatment groups. (**E**) Differential metabolite correlation plot. The highest correlation is 1, representing complete positive correlation (in red), while the lowest correlation is −1, representing complete negative correlation (in blue). Areas without color indicate that the calculated correlation is not statistically significant at the specified threshold. The plot displays the correlation of the top 20 differential metabolites. Among them, A represents 9,10–Epoxyoctadecanoic acid, B stands for 11Z–hexadecenoic acid, C denotes carprazidil, D is for stearic acid, E corresponds to secologanin, F is DG(8:0/10:0/0:0), G signifies 13′–Hydroxy–alpha–tocopherol, H is 5–(3′–Carboxy–3′–oxopropenyl)–4,6–dihydroxypicolinate, I is tridecanoic acid (tridecylic acid), J indicates 1–Palmitoylglycerol, K refers to 4alpha–hydroxymethyl–ergosta–7,24(241)–dien–3beta–ol, L stands for 3–hydroxypristanic acid, M represents 3–Methyldioxyindole, N denotes MG(16:1(9Z)/0:0/0:0), O signifies Ile Arg Glu, P is UDP–N–acetylmuramoyl–L–alanine, Q represents dynorphin B (6–9), R stands for Phe–Pro–Ile, S indicates spisulosine, and T corresponds to spermine.

**Figure 2 animals-14-01045-f002:**
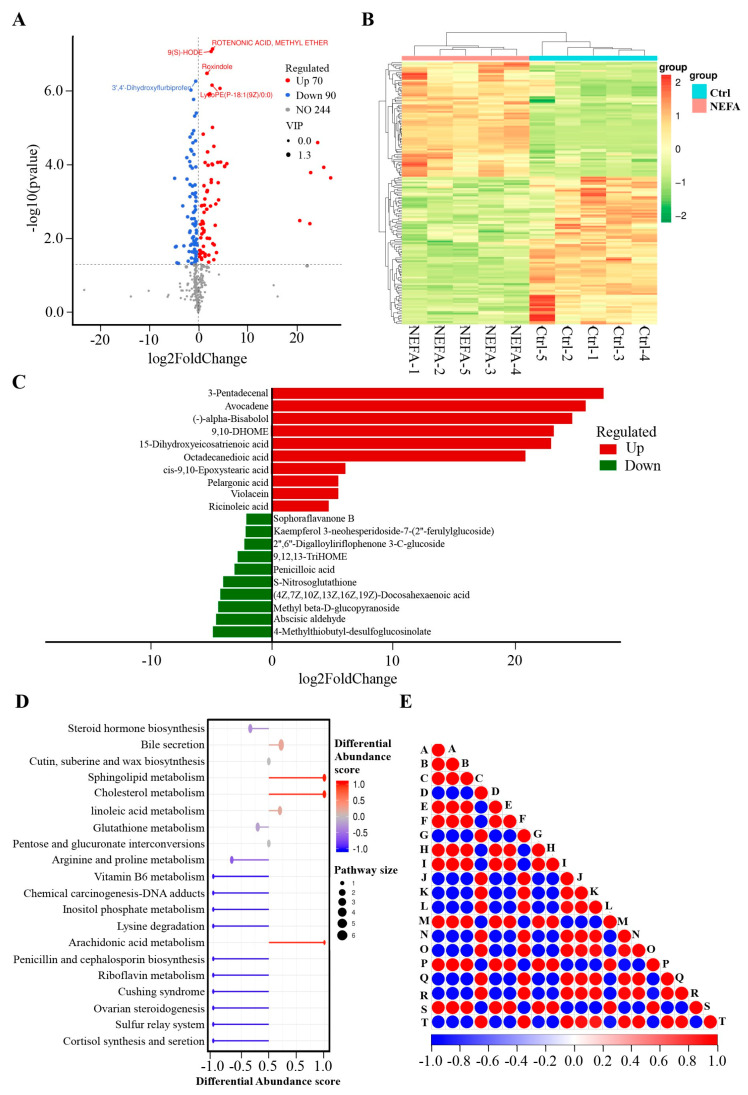
Effect of high NEFA levels on the metabolomics of BEMCs in the negative model. (**A**) Volcano of metabolites with significant differences between control and NEFA treatment groups. The criteria for upregulated differential metabolites should be a VIP > 1, *p* < 0.05, and log2(Foldchange) > 0. Conversely, the criteria for downregulated differential metabolites should be a VIP > 1, *p* < 0.05, and log2(Foldchange) < 0. (**B**) The clustering heat map of differential metabolites between control and NEFA treatment groups. (**C**) Top 10 elevated and reduced differential metabolites identified between the control and NEFA groups in the negative model. (**D**) KEGG pathways enriched by differential metabolites between control and NEFA treatment groups. (**E**) Differential metabolite correlation plot. The highest correlation is 1, representing complete positive correlation (in red), while the lowest correlation is −1, representing complete negative correlation (in blue). Areas without color indicate that the calculated correlation is not statistically significant at the specified threshold. The plot displays the correlation of the top 20 differential metabolites. Among them, A represents rotenonic acid and methyl ether, B stands for 9(S)–HODE, C denotes roxindole, D is for 3′,4′–Dihydroxyflurbiprofen, E corresponds to LysoPE(P–18:1(9Z)/0:0), F is deoxycholic acid, G signifies hydroxyproline, H is 8–Amino–8–demethylriboflavin 5′–phosphate, I is glutathionylspermidine, J indicates (1R,6R)–1,4,5,5a,6,9–Hexahydrophenazine–1,6–dicarboxylate, K refers to 2,6–Dihydroxy–N–methylmyosmine, L stands for deacetylisoipecoside, M represents 6‴–Deamino–6‴–hydroxyneomycin C, N denotes perfluoro(methylcyclohexane), O signifies PE(18_1(9Z)_0_0), P is PE(18:1(9Z)/0:0), Q represents p–Tolyl Sulfate, R stands for salfredin B11, S indicates (–)–alpha–Bisabolol, and T corresponds to tyrosyl–methionine.

**Figure 3 animals-14-01045-f003:**
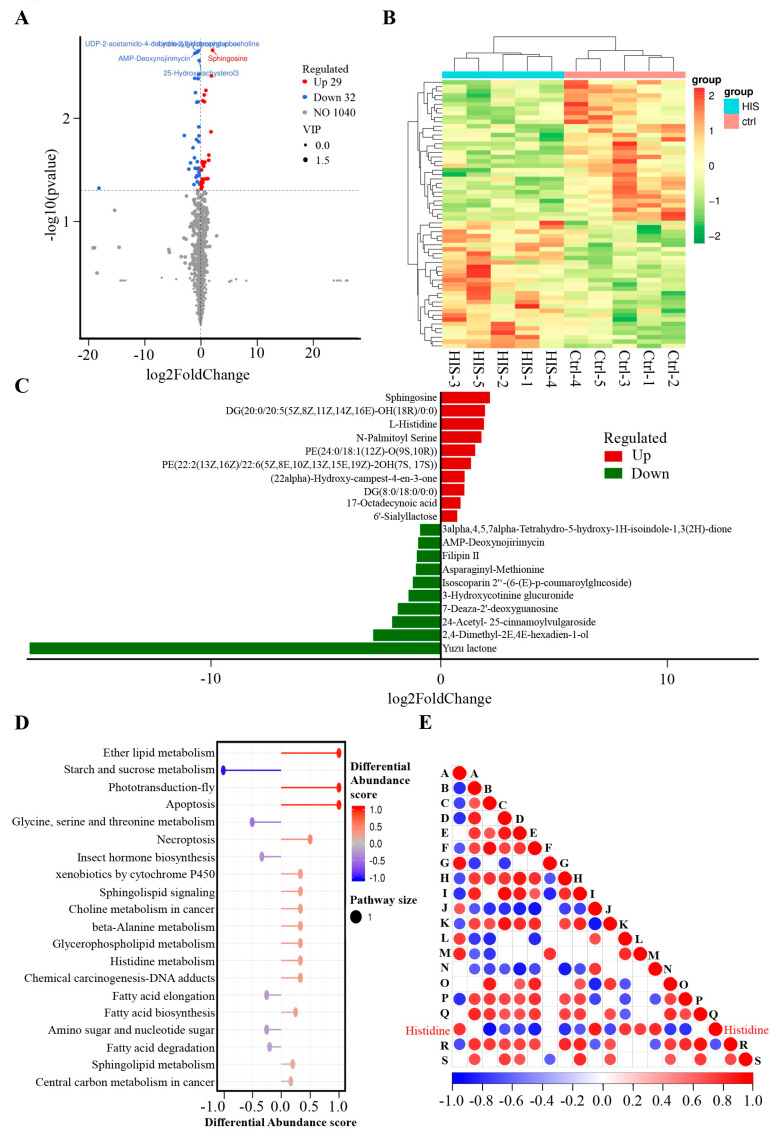
Effect of L-histidine on the metabolomics of BEMCs in the positive model. (**A**) Volcano of metabolites with significant differences between control and L-histidine treatment groups. The criteria for upregulated differential metabolites should be a VIP > 1, *p* < 0.05, and log2(Foldchange) > 0. Conversely, the criteria for downregulated differential metabolites should be a VIP > 1, *p* < 0.05, and log2(Foldchange) < 0. (**B**) The clustering heat map of differential metabolites between control and L-histidine treatment groups. (**C**) Top 10 elevated and reduced differential metabolites identified between the control and L-histidine groups in the positive model. (**D**) KEGG pathways enriched by differential metabolites between control and L-histidine treatment groups. (**E**) Differential metabolite correlation plot. The highest correlation is 1, representing complete positive correlation (in red), while the lowest correlation is −1, representing complete negative correlation (in blue). Areas without color indicate that the calculated correlation is not statistically significant at the specified threshold. The plot displays the correlation of the top 20 differential metabolites. Among them, A represents sphingosine, B stands for 1–Linoleoylglycerophosphocholine, C denotes UDP–2–acetamido–4–dehydro–2,6–dideoxyglucose, D is for AMP–Deoxynojirimycin, E corresponds to 25–Hydroxytachysterol3, F is etrimfos, G signifies DG(20:0/20:5(5Z,8Z,11Z,14Z,16E)–OH(18R)/0:0), H is filipin II, I is creatine, J indicates DG(8:0/18:0/0:0), K refers to trans–Dec–2–enoyl–CoA, L stands for Cer(d18:0/20:3(8Z,11Z,14Z)–2OH(5,6)), M represents PE(22:2(13Z,16Z)/18:3(10,12,15)–OH(9)), N denotes 6′–Sialyllactose, O signifies levanbiose, P is cyanthoate, Q represents pyroglutamine, L–histidine stands for L–histidine, R indicates 2,4–Dimethyl–2E,4E–hexadien–1–ol, and S corresponds to norpandamarilactonine A.

**Figure 4 animals-14-01045-f004:**
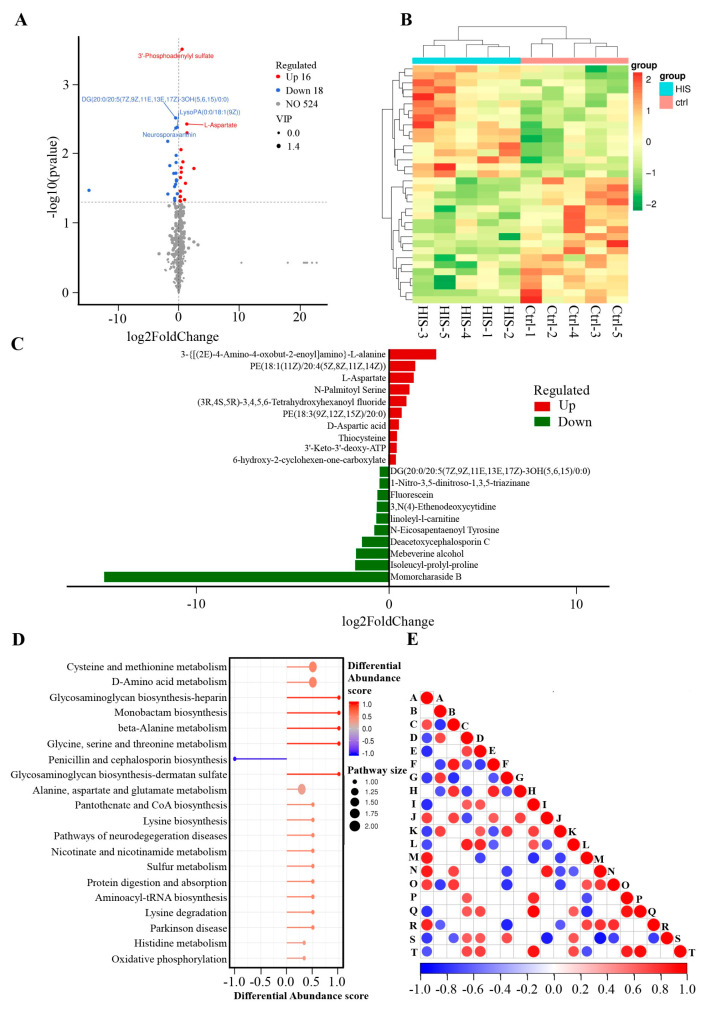
Effect of L-histidine on the metabolomics of BEMCs in the negative model. (**A**) Volcano of metabolites with significant differences between control and L-histidine treatment groups. The criteria for upregulated differential metabolites should be a VIP > 1, *p* < 0.05, and log2(Foldchange) > 0. Conversely, the criteria for downregulated differential metabolites should be a VIP > 1, *p* < 0.05, and log2(Foldchange) < 0. (**B**) The clustering heat map of differential metabolites between control and L-histidine treatment groups. (**C**) Top 10 elevated and reduced differential metabolites identified between the control and L-histidine groups in the negative model. (**D**) KEGG pathways enriched by differential metabolites between control and L-histidine treatment groups. (**E**) Differential metabolite correlation plot. The highest correlation is 1, representing complete positive correlation (in red), while the lowest correlation is −1, representing complete negative correlation (in blue). Areas without color indicate that the calculated correlation is not statistically significant at the specified threshold. The plot displays the correlation of the top 20 differential metabolites. Among them, A represents 3′–Phosphoadenylyl sulfate, B stands for DG(20:0/20:5(7Z,9Z,11E,13E,17Z)–3OH(5,6,15)/0:0), C denotes L–Aspartate, D is for LysoPA(0:0/18:1(9Z)), E corresponds to neurosporaxanthin, F is PE(18:1(11Z)/20:4(5Z,8Z,11Z,14Z)), G signifies Isoleucyl–prolyl–proline, H is 6–hydroxy–2–cyclohexen–one–carboxylate, I is PC(17:2(9Z,12Z)/0:0), J indicates D–aspartic acid, K refers to (1R)–5–[2–[(1S,7As)–1–[(1S)–1–(3–hydroxy–3–methylbutoxy)ethyl]–7a–methyl–2,3,3a,5,6,7–hexahydro–1H–inden–4–ylidene]ethylidene]–4–methylidenecyclohexane–1,3–diol, L stands for deacetoxycephalosporin, M represents 3′–Keto–3′–deoxy–ATP, N denotes 3–{[(2E)–4–Amino–4–oxobut–2–enoyl]amino}–L–alanine, O signifies thiocysteine, P is (5alpha)–23–Methyl–4–aza–21–norchol–1–ene–3,20–dione, Q represents N–Eicosapentaenoyl tyrosine, R stands for O–acetyl–ADP–ribose, S indicates 2′–Deoxyinosine 5′–phosphate, and T corresponds to tenofovir exalidex.

**Figure 5 animals-14-01045-f005:**
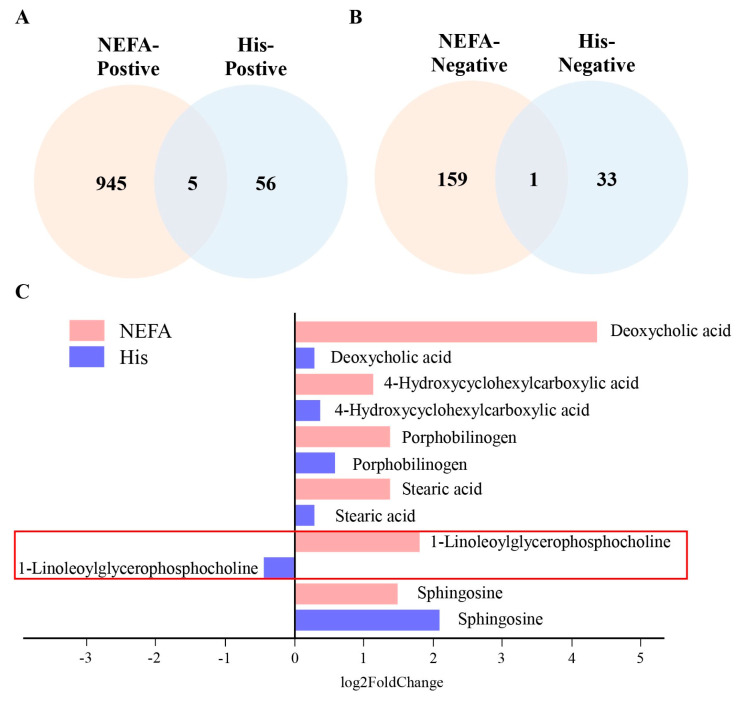
Venn diagram of differential metabolites. (**A**,**B**) The Venn diagrams highlighting the differential metabolites associated with NEFA and histidine in the positive and negative models, respectively. (**C**) The bar chart illustrates the expression differences of the overlapping metabolites.

## Data Availability

The data underlying this article will be shared on reasonable request to the corresponding author.

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
