# Peer review of "Comparative Metabolomic Profiling of L-Histidine and NEFA Treatments in Bovine Mammary Epithelial Cells"

_animals, 2024, doi:10.3390/ani14071045_

Round 1
Reviewer 1 Report
Comments and Suggestions for Authors
In this manuscript, the authors have conducted comparative metabolomic profiling of L-histidine and NEFA treatments in bovine mammary epithelial cells. The overall logic of this manuscript is relatively clear. However, I think the authors' work is not sufficient to support the conclusion of this manuscript “Our findings underscore the complexity of BMECs responses to NEFA and highlight the potential of L-histidine in ameliorating the adverse effects of high NEFA levels....”. Furthermore, while I appreciate the effort of the work presented, I would like the authors to consider the following correction.
1. In the introduction, some genes such as ACSL, ACADL, CPT 1A, and ACC, appear for the first time, which should be given their full names, with abbreviations noted in parentheses.
2. In the materials and methods section, how were the concentrations of NEFA and L-histidine determined?
3. After NEFA or L-histidine treatment, was cell viability affected? Please supplement the data.
4. Differential Metabolite Correlation Plot in Figure 1 and 2 did not show obvious color alterations. Are all correlation coefficients 1 or -1? Please provide raw data including correlation coefficient.
5. The logic of discussion section is confused. Please reorganize and improve it.
6. In abstract, the author mentioned that “Our findings underscore the complexity of BMECs responses to NEFA and highlight the potential of L-histidine in ameliorating the adverse effects of high NEFA levels...”, which was not appropriate. Because the co-treatment of NEFA and L-histidine was not conducted to investigate the effect of L-histidine on high NEFA adverse reactions.
Author Response
Thank you very much for giving us an opportunity to submit a revised version of the manuscript, animals-2909377. We wanted to thank all the reviewers for their insightful comments. We were able to address all the comments with experiments if applicable. We have included a detailed point-by-point response to the reviewers' comments as an attachment to this correspondence.

Reviewer 2 Report
Comments and Suggestions for Authors
This study aims to delve deeper into the metabolomic responses of BMECs to high NEFA levels and to assess the potential of L-histidine in counteracting these effects.
The authors have tackled a very interesting and useful topic and the manuscript is written well. I enjoyed particularly the excellent visualization approach and the nice graphs which help to summarize the results correctly and interestingly. The authors have applied advanced methodologies and have produced a lot of data. The data have not been fully analyzed, but these are adequate for adding in the current manuscript. There is potential for a second high-quality manuscript, based on further analysis of existing data.
I have only some minor points that should be addressed in the Discussion.
1. The authors have missed 2 or 3 recent references (after December 2023), which is OK, but possibly they can search and include them in the Discussion.
2. The authors should add a paragraph about the clinical significance of their findings and how in the future we can ask all cattle farmers to perform this excellent testing to assess the metabolic profiling of the mammary gland cells of their cows.
3. A sentence regarding the commercialization of the findings will be helpful. Is there a patent pending? No need to write details, these can be addressed in a further paper, but a brief mention (one sentence) is really worthy.
All in all, an excellent manuscript, which surely deserves publication.
Author Response

(The authors gave the same response as above.)

Reviewer 3 Report
Comments and Suggestions for Authors
This is an interesting work. The authors follow the common issue that has been concerned by the dairy industry. Also, this work has identified some metabolites related to the L-Histidine and NEFA treatments in BMECs. Nevertheless, there are some concerns that need to be addressed as follows:
1. Line 39-40: this work is lacking in addressing the interaction between L-Histidine and NEFA, which cannot support the current point. please rewrite this.
2. Line 95-96: please explain the rationale for choosing the current concentration and time of L-Histidine and NEFA treatment.
3. why do you only select 20 metabolites for correlation analysis? Please explain it.
4. please define the criteria for upregulated and downregulated? Upregulated: VIP>1 and -log10(pvalue) >1.3?
5.Line50: what does the acronym ACC stand for?
6. Some software needs to be cited?
Comments on the Quality of English Language
This is an interesting work. The authors follow the common issue that has been concerned by the dairy industry. Also, this work has identified some metabolites related to the L-Histidine and NEFA treatments in BMECs. Nevertheless, there are some concerns that need to be addressed as follows:
1. Line 39-40: this work is lacking in addressing the interaction between L-Histidine and NEFA, which cannot support the current point. please rewrite this.
2. Line 95-96: please explain the rationale for choosing the current concentration and time of L-Histidine and NEFA treatment.
3. why do you only select 20 metabolites for correlation analysis? Please explain it.
4. please define the criteria for upregulated and downregulated? Upregulated: VIP>1 and -log10(pvalue) >1.3?
5.Line50: what does the acronym ACC stand for?
6. Some software needs to be cited?
Author Response

(The authors gave the same response as above.)

Reviewer 4 Report
Comments and Suggestions for Authors
Comparative Metabolomic Profiling of L-Histidine and NEFA Treatments in Bovine Mammary Epithelial Cells
Review
This is a detailed experimental investigation with abundant parameters indicating the mechanisms of L histidine influence on metabolic processes in mammary epithelial cells in different treatments with NEFA. NEFA significantly disrupts the metabolic processes in the cells of the mammary gland, and L histidine has a modulatory effect leading to the rebalancing of disturbed metabolic relations. This is interesting manuscript about relation between L-histidin and NEFA pathways in mammary gland – on Bovine Mammary Epithelial Cells.
Histidine belongs to Group 1 AA, for which post-liver supply closely matches mammary net uptake, with the latter balancing output in milk protein net synthesized within the mammary gland. Problematically, recommendations for His requirement have varied considerably from 2.4 to 3.2% of MP supply. These estimates have been based on the relation of MPY to the proportion of His in MP supply: His requirement was determined at optimal MPY. Unfortunately, this approach does not provide any information on the mechanisms that underlie the variation in efficiency of His utilization under different nutritional conditions. The determination of potential mechanisms involved in the variations in the efficiency of His utilization would help refine His optimal supply. Under limited His supply, these mechanisms would include reduced catabolism, more efficient mammary usage, and use of His labile pools. Shifts in partition between anabolic and catabolic fates across tissues in response to lower His supply would involve decreased net liver removal, as observed when AA supply was decreased in dairy cows, and increased mammary blood flow with improved fractional mammary extraction, as observed under limited His supply in dairy goats. All of the above represent assumptions about the importance of L histidine. However, in this paper, the functions of the mentioned amino acid and the metabolic processes related to NEFA are precisely matched. This paper largely resolves the issue of metabolic flows and regulation of these two metabolites. Relations were established at the level of metabolomic analysis.
General note – Ketosis and ketone and NEFA are not same. Ketosis occurs when there is an increased production of ketone bodies. Ketone bodies are not fats. Ketone bodies are water-soluble molecules or compounds that contain the ketone groups produced from fatty acids (NEFA) by the liver (ketogenesis). Ketone bodies are produced by the liver during periods of caloric restriction of various scenarios: low food intake (fasting), carbohydrate restrictive diets, starvation…Fats stored in adipose tissue are released from the fat cells into the blood as free fatty acids and glycerol when insulin levels are low and glucagon and epinephrine levels in the blood are high. This occurs between meals, during fasting, starvation and strenuous exercise, when blood glucose levels are likely to fall. Fatty acids are very high energy fuels and are taken up by all metabolizing cells that have mitochondria. This is because fatty acids can only be metabolized in the mitochondria. Red blood cells do not contain mitochondria and are therefore entirely dependent on anaerobic glycolysis for their energy requirements. In all other tissues, the fatty acids that enter the metabolizing cells are combined with coenzyme A to form acyl-CoA chains. These are transferred into the mitochondria of the cells, where they are broken down into acetyl-CoA units by a sequence of reactions known as β-oxidation. The acetyl-CoA produced by β-oxidation enters the citric acid cycle in the mitochondrion by combining with oxaloacetate to form citrate. This results in the complete combustion of the acetyl group of acetyl-CoA (see diagram above, on the right) to CO2 and water. The energy released in this process is captured in the form of 1 GTP and 9 ATP molecules per acetyl group (or acetic acid molecule) oxidized.
Ketosis, therefore, is a decompensated state that results from increased use of fat for energy purposes when ketone bodies are created. Ketone bodies are not fatty, they are not a type of fat, and therefore, even though there is a functional and pathogenetic connection with NEFA, they cannot be identified. Take care throughout the text about this. For example, there is already one incorrect sentence in Line 15. Cows can have high levels of NEFA but not have extensive ketogenesis and ketosis develop. Conversely, ketosis can occur without having very high levels of NEFA in the cows' blood.
Good definition in line 46 and 47.
Already in line 49 and 50 there are abbreviations that need to be explained at the first appearance. Consider a list of abbreviations at the end of the paper if the form of the template allows it.
The introduction is well and concisely written, with a clear emphasis on what is known so far and what is not known, and will be explored in this paper.
Line 79 – “Our preliminary findings” - if you have published the results, cite the paper or put unpublished.
M and M - Cell culture and Treatments You need to define the positive and negative model in detail. The Cell culture and Treatments chapter needs to be written in much more detail. Acknowledgments to whoever donated the cell line should be in the Acknowledgments section, not in M and M. Was treatment with one or more concentrations of NEFA, with one or more concentrations of L histidine, and were there differences in the length of exposure?
Results – Line 146 - impact of high levels of NEFA… Where is the impact of low or medium concentrations seen? Please have a uniform style of describing groups and treatments in the M and M and Results section.
The results are clear and readable. The charts are well executed and unambiguous. However, please describe in detail the models in M and M exactly as you named them in the subheadings of the Results.
The discussion is correctly written, but it is necessary to discuss the results given in line 284/285 in much more detail.
The conclusions are correct and in accordance with the hypotheses and the obtained results.
Supplementary Figures 1-4 can also become part of the main manuscript.
Make sure all references are written according to MDPI style.
Author Response

(The authors gave the same response as above.)

Round 2
Reviewer 3 Report
Comments and Suggestions for Authors
The criteria for upregulated differential metabolites should be the VIP>1, P<0.05, and log2(Foldchange)>0. while the criteria for downregulated differential metabolites should be the VIP>1, P<0.05, and log2(Foldchange)< 0. please add this in the manuscript.
Comments on the Quality of English LanguageNo comments
Author Response
Dear Reviewer,
Thank you for your constructive comments and suggestions regarding the criteria for defining differential metabolites in our manuscript. We appreciate the time you have taken to review our work and your insightful feedback. As per your suggestion, we have now included the specific criteria for the identification of upregulated and downregulated differential metabolites in our study. Please find the revised sections highlighted in the manuscript for your convenience.
Thank you once again for your valuable feedback.
Sincerely,
Songjia Lai